# Effect of caffeine citrate on diaphragmatic electrical activity in pre-term newborns

**Tatiana B.B. Zidan** [1]*, **Eduardo J. Troster**[1], **Jennifer Beck**[2], **Livia R. Sanches**[3], **Carlos Eduardo S. Ferreira**[3], **Romy S.B. Zacharias**[1], **Celso M. Rebello**[1]

**1** Departamento Materno Infantil, Hospital Israelita Albert Einstein, São Paulo, São Paulo, Brazil, **2** Department of Pediatrics, University of Toronto, Toronto, Ontario, Canada, **3** Laboratório Clínico, Hospital Israelita Albert Einstein, São Paulo, São Paulo, Brazil

\* tatizidan@hotmail.com

## Abstract

The gold standard treatment for apnea of prematurity is caffeine citrate, which is known for its effect on diaphragm muscle activity. The purpose of this study was to investigate the electrical activity of the diaphragm in preterm newborns, which was measured 30 minutes before and 60 minutes after the administration of a loading dose of caffeine citrate. In this prospective, observational, longitudinal study at a tertiary-level neonatal ICU, data were collected from 36 patients (13 females, 23 males) with a mean gestational age of 31 2/7 ± 2 1/7 weeks and a mean birth weight of 1532 ± 439 grams. The average caffeine level was 15.7 ± 5.9 mg/dL (reference range: 5–30 mg/dL). The results revealed significant increases in all the parameters used to determine the inspiratory phases of the respiratory cycle postcaffeine administration (p < 0.001). The study concluded that caffeine administration significantly enhances diaphragmatic electrical activity in preterm newborns.

## Introduction

Prematurity and the survival of preterm newborns have increased in recent years. The World Health Organization estimates that 15 million premature births occur each year. The worldwide rate of prematurity ranges from 5 to 18%, and its complications are the main cause of mortality in children under 5 years of age [1,2].

Apnea of prematurity is one of the most frequent diagnoses in neonatal intensive care units; its incidence is inversely proportional to the newborn's gestational age, and it affects almost all newborns under 28 weeks of gestational age. Recurrent episodes of apnea, bradycardia and hypoxemia worsen the prognosis and increase morbidity among this population [3].

The gold standard treatment for apnea of prematurity is caffeine citrate, which is the most frequently used drug in neonatal intensive care; its routine use is widely recommended for preterm newborns [4]. Among its benefits are reductions in the occurrence of apnea during prematurity, extubation failure, the duration of mechanical ventilation and oxygen need, and the incidence of bronchopulmonary dysplasia [5–7]. Long-term studies have shown reduced neurodevelopmental disability at 18–21 months with caffeine citrate use, with follow-ups at 5 and 11 years showing improved gross motor function and reduced motor impairment [8–10].

**Data availability statement:** All relevant data are available from Figshare repository (DOI: https://doi.org/10.6084/m9.figshare.26752516).

**Funding:** The author(s) received no specific funding for this work.

**Competing interests:** The study was designed and conducted at the Hospital Israelita Albert Einstein in São Paulo, and none of the authors have any conflicts of interest related to patents. The analysis of the collected data was conducted in Toronto by Dr. Beck's team. We confirm that Dr. Beck's participation in the study did not involve any conflicts of interest.

Caffeine citrate acts by blocking A1 and A2 adenosine receptors, stimulating the respiratory center [11]; it also increases diaphragmatic electrical activity and contractility, although the exact mechanism remains unknown [12,13].

Diaphragmatic electrical activity (Edi) can be monitored, and this parameter is used to adjust mechanical ventilation in neurally adjusted ventilatory assist mode (NAVA – neurally adjusted ventilatory assist) [14]. In this ventilatory strategy, the inspiratory pressure is proportional to the intensity of Edi, and ventilator pressurization ceases when the diaphragm's neural activity begins to decline. The maximum Edi value (Edi max) indicates inspiratory nerve stimulus intensity and breath duration, whereas the minimum Edi (Edi min) represents diaphragm basal tone, preventing alveolar collapse during expiration.

Few studies have investigated the effect of caffeine on the electrical activity of the diaphragm in preterm newborns [15–17]. The aim of this study was to analyze Edi behavior in preterm newborns before and after the administration of a loading dose of caffeine citrate. The secondary objectives were to observe the respiratory rate and occurrence of breathing pauses after drug administration.

We hypothesized that Edi would increase after drug administration.

## Materials and methods

### Study design

This was a prospective, observational, longitudinal study (pre- and postintervention) conducted at a tertiary-level neonatal ICU.

### Inclusion criteria

Newborns of both genders, of any chronological age, with a gestational age of less than 37 weeks and clinical indications for caffeine citrate treatment, were included. The criteria included preterm newborns with a gestational age of ≤28 weeks, those between 28 and 32 weeks receiving mechanical ventilation, and those with a clinical diagnosis of apnea. Gestational age was determined by the date of the mother's last menstrual period or early first-trimester ultrasound in its absence. Central apnea was clinically defined as breathing pauses ≥20 seconds, or shorter with bradycardia (heart rate <100 bpm), cyanosis, or pallor. Newborns were included if they had ≥2 clinical apnea episodes within an hour. Caffeine was started prophylactically for all patients ≤28 weeks and those 28–32 weeks under mechanical ventilation or therapeutically for all patients with a clinical apnea diagnosis.

### Exclusion criteria

Patients with major congenital malformations, upper digestive tract anomalies preventing esophageal catheter insertion, primary neurological or drug-related abnormalities impairing respiratory center function, or those under high-frequency mechanical ventilation were excluded.

### Patient population

We included patients on invasive mechanical ventilation, namely, a neutrally adjusted ventilatory assist (NAVA), synchronized intermittent mandatory ventilation (SIMV) and assisted control (AC), as well as those on noninvasive ventilation (noninvasive NAVA and CPAP), and those breathing spontaneously. Respiratory support was defined based on clinical parameters, with no changes between baseline and postcaffeine measurements. None of the patients were sedated.

Out of the 78 eligible patients, 42 were excluded (4 major malformations, 18 refusal of consent, 7 data recording issues, 1 severe asphyxia and 12 concurrent use of the system), resulting in 36 included patients.

Patients were recruited between July 19, 2018, and October 3, 2019.

## Caffeine treatment

The caffeine citrate loading dose was 20 mg/kg, as recommended by the largest randomized controlled trial and the US Food and Drug Administration [7,18,19].

## Evaluation of diaphragmatic electrical activity

A naso/oro gastric feeding tube designed to measure the electrical activity of the diaphragm [Maquet, Getinge Group, Solna, Sweden] was inserted and positioned in the lower esophagus, close to the crural diaphragm. The catheter contained nine electrodes that captured and processed the Edi into a waveform and removed contamination from the electrical signals from the heart, the esophagus and the environment [14,20,21]. The Edi and all the mechanical ventilator parameters were continuously recorded and transferred simultaneously to a computer via Servo-Tracker software [Maquet, Getinge Group, Solna, Sweden] and using a data transmission cable adaptable to a Servo-$n$ mechanical ventilator [Maquet, Getinge Group, Solna, Sweden]. The data were collected 30 minutes before and 60 minutes after the caffeine loading dose (Fig. 1, see blue bars).

## Edi signal analysis

The Edi waveform was automatically analyzed automatically (by Servo-n or Servo-I software, Maquet Getinge Group, Sweden) and was characterized by its timing and amplitude. The parameters of interest included the peak Edi amplitude, end-expiratory amplitude, and neural respiratory rate, which were quantified before and after caffeine administration. To define an "Edi breath", three specific time points of the neural respiratory cycle were determined: beginning of neural inspiration (beginning of Edi increase defined as greater than 0.5 μV above the basal line (IOn), peak of Edi (highest Edi value during inspiration "PeakIon"), and end of inspiration was determined by a decrease equal to or greater than 70% of the Edi (PIOff) (Fig. 2). Additionally, the end expiratory Edi (lowest Edi value during expiration) was determined.

Neural respiratory activity was recorded for the inspiratory and expiratory phases (Fig. 3). Respiratory pauses were quantified based on flat Edi periods of 5–10 sec, 10–19 sec or longer.

## Serum caffeine levels

Serum caffeine levels were measured 24 hours after the loading dose or maintenance dose, according to scheduled routine collections. No venous punctures were performed exclusively for this purpose.

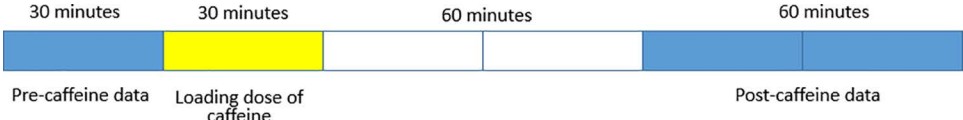

**Fig. 1. Schematic representation of the study design.** Precaffeine data: first blue column. Loading dose of caffeine: yellow column. Time after loading: white column. Postcaffeine data: second blue column.

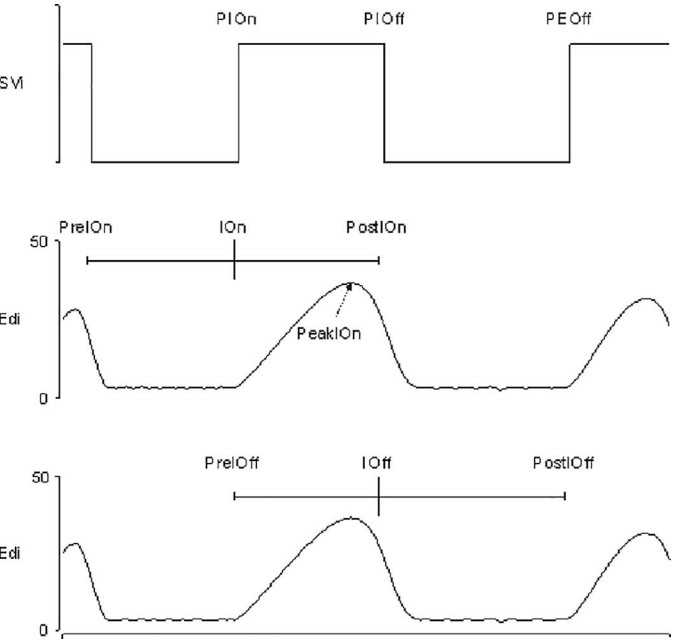

**Fig. 2. Schematic representation of the methodology used to collect data on the electrical activity of the diaphragm for all newborns.** Svi: pneumatic state; Edi: diaphragmatic electrical activity; PIOn: start of pneumatic inspiration; PIOff: start of pneumatic expiration; PEOff: start of the next inspiration cycle; PreIOn: previous inspiratory stop (70% peak Edi); IOn: inspiratory start (first sample increase of a detected Edi breath); PostIOn: inspiratory stop (70% peak Edi); PeakIOn: inspiratory peak Edi; PreIOff: inspiratory start; IOff: inspiratory stop; PostIOff: next inspiratory start.

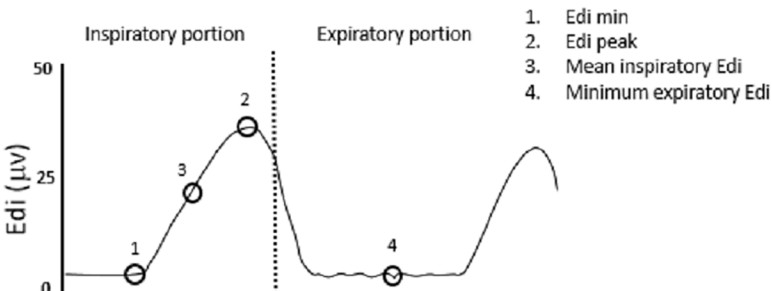

**Fig. 3. Schematic representation of the methodology for analyzing data on the electrical activity of the diaphragm in the inspiratory and expiratory phases.** 1. Edi min 2. Edi peak 3. Mean inspiratory Edi 4. Minimum expiratory Edi (MinEEdi).

A 400-µL blood sample was collected, and caffeine levels were analyzed via a method developed at the Laboratory of Clinical Chemistry of the Hospital Israelita Albert Einstein. This analysis was performed via ultrahigh-performance liquid chromatography with a diode array detector (UHPLC-DAD). Chromatographic separation was performed via a C18 column (2.1x 50 mm, 1.8 m); the mobile phase consisted of water and methanol, and the flow rate was 0.4 mL/minute. The sample preparation involved protein precipitation with methanol and zinc sulfate, followed by filtration and injection of the supernatant via UHPLC-DAD equipment [22,23].

## Statistics

The sample calculation was based on a convenience sample collected between July 2018 and October 2019, including all newborns who met the criteria for inclusion, within the data collection period.

Parameter distributions were studied via histograms and boxplots, and generalized estimating equation models were applied for the inferential analysis [24], considering the correlation between measurements from the same newborn before and after caffeine administration. Normal or gamma distributions with a logarithmic link function were used, with the best-fitting distribution chosen based on QIC goodness-of-fit results [25].

Comparisons between pre- and postcaffeine measurements in the total sample were performed, and the model fit considered the effect of evaluation time. To compare the measurements within the pre- and postcaffeine groups, the mode and time of the evaluation and the interaction effect between mode and time were used as the explanatory variables of the model. The p values obtained in the multiple comparisons between the evaluation times within each mode were corrected via the sequential Bonferroni method.

The analyses were performed via SPSS statistical software [26], and a significance level of 0.05 was adopted.

## Ethical considerations

The study was approved by the Hospital Israelita Albert Einstein Ethics Committee (approval number 2.730.352). Data were collected after informed consent (verbal and written) was obtained from parents.

The study was registered internally at our hospital on two platforms: the SGPP (General System of Research Protocols) and the hospital's Ethics and Research Committee. Research projects are subsequently audited by the hospital's scientific research integrity office. It was also registered and approved in the Brazilian Registry of Clinical Trials, a body that is part of the World Health Organization.

## Results

Data were collected from 43 patients, with seven losses due to data recording problems, including problems with the data transmission cable and incompletely captured data, resulting in 36 analyzed patients. The demographic characteristics and types of respiratory support used are shown in Table 1. The caffeine level was 15.7 ± 5.9 mg/dl (reference range: 5–30 mg/dl). The minimum value obtained was 4.8 mg/dl, and the maximum value was 31.8 mg/dl.

Caffeine was administered therapeutically in 20 patients (55.5%) and prophylactically in 16 patients (44.5%). Data were collected in the first 12 hours of life for 25 patients (69.4%), 3 patients (8.3%) had their data collected between 12 and 24 hours of life, and 8 patients (22.3%) were monitored for more than 24 hours of life. The average age at data collection was 2 hours of life.

The results of the comparisons of measurements before and after caffeine citrate administration are shown in Table 2. We evaluated the minimum inspiratory Edi (MinIEdi), maximum inspiratory Edi (MaxIEdi), mean inspiratory Edi and MinEEdi before and after the caffeine citrate loading dose. MinEdi increased from 2,32–2,93 μV (p =0.007), and MaxEdi increased from 10,35–12,17 μV (p<0.05). The mean inspiratory Edi increased from 6,76–8,11 μV (p<0.05), and MinEEdi increased from 2,31–2,98 μV (p=0.004). We also evaluated the neural respiratory rate before and after caffeine administration. There was a statistically significant increase in the neural respiratory rate from 66.5 to 75.9 bpm (breaths per minute) (p=0.01).

**Table 1. Demographic characteristics of the studied population and types of ventilatory support.**

|  | Value |
|---|---|
| Birth weight (g) - mean±sd | 1532 ± 439 |
| Min (g) | 580 |
| Max (g) | 2845 |
| Gestational age (weeks) - mean±sd | 31.1 ± 2.1 |
| Min (weeks) | 25 |
| Max (weeks) | 35.1 |
| Female N (%) | 13 (36.1) |
| Ventilatory support |  |
| None N (%) | 7 (19.5) |
| CPAP N (%) | 11 (30.5) |
| NIV NAVA N (%) | 13 (36.1) |
| Invasive Mechanical ventilation N (%) | 5 (13.9) |
| Antenatal corticosteroids N (%) | 18 (50) |

sd, standard deviation; NIV NAVA, non invasive NAVA.

**Table 2. Estimated means (95% confidence intervals) and results of comparisons between pre and post caffeine administration in preterm newborns.**

| Values | Evaluation | | p-values |
|---|---|---|---|
|  | **Pre-caffeine** | **Post-caffeine** |  |
| Minimum inspiratory Edi (µV) | 2.32 (1.85: 2.91) | 2.93 (2.39: 3.59) | **0.007** |
| Maximum inspiratory Edi (µV) | 10.35 (8.11: 13.21) | 12.17 (9.93: 14.93) | **0.037** |
| Mean Edi between inspiratory onset and end (µV) | 6.76 (5.39: 8.46) | 8.11 (6.70: 9.81) | **0.011** |
| Minimum expiratory Edi (µV) | 2.31 (1.83: 2.92) | 2.98 (2.42: 3.66) | **0.004** |
| Inspiratory peak Edi (ms) | 1729 (1482.3: 2018) | 2907 (2601: 3248) | **<0.001** |
| RR (bpm) | 66.5 (61,6 - 71,7) | 75.9 (68,7 - 83,8) | 0.001 |

p-values<0,05 highlighted in bold; values expressed as estimated means and 95% confidence intervals; Edi: diaphragmatic electrical activity; ms: milliseconds, RR: respiratory rate

We evaluated the number of respiratory pauses and apnea episodes before and after the caffeine citrate loading dose. There was a reduction in the number of respiratory pauses between 5–10 s and 11–19 s after the loading dose; however, this reduction was not statistically significant (Fig. 4).

## Discussion

This study demonstrated that caffeine administration in preterm newborns promotes an increase in the electrical activity of the diaphragm.

Our study is the first to document the serum caffeine level and to demonstrate adequate therapeutic levels in newborns whose diaphragmatic electrical activity data were collected. These data are useful for ensuring that the serum concentration of the drug is adequate.

The effect of caffeine on the central nervous system in preterm newborns is well known and documented. Caffeine acts by inhibiting adenosine receptors, which stimulate the respiratory center, improve responsiveness to increased $CO_2$ and reduce neuronal apoptosis in hypoxic-ischemic lesions [11,27]. There are, however, few studies on its effects on diaphragm function.

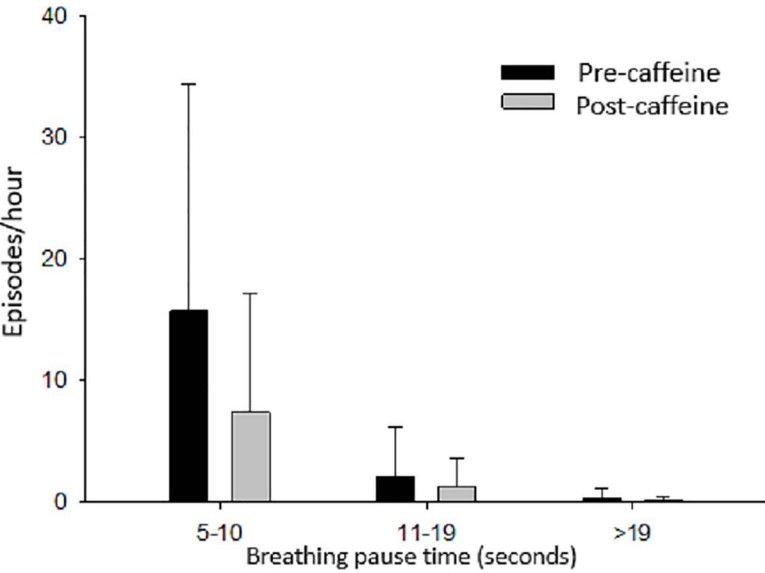

**Fig 4. Schematic representation of the number of respiratory pauses pre- and postcaffeine loading dose, results as mean ± standart deviation.** Precaffeine data: black columns Postcaffeine data: gray columns.

Our study revealed an increase in diaphragm electrical activity in all phases of the respiratory cycle (inspiration and expiration) after caffeine loading. All recorded respiratory cycles were studied separately. The Edi values at the beginning, end, peak, minimum and mean of the inspiratory and expiratory cycles significantly increased. Additionally, there was an increase in the respiratory rate after the loading dose was administered.

These data are consistent with those of previous studies showing an improvement in the contractility of respiratory muscles [12,28] and the action of caffeine citrate on peripheral chemoreceptors [29]. It has also been shown that caffeine causes increases in minute volume and tidal volume, a decrease in resistance and an increase in lung compliance in newborns diagnosed with bronchopulmonary dysplasia [30]. Dekker et al. also demonstrated an increase in respiratory effort with increased minute volume and tidal volume [31]. These findings may be explained by an improvement in respiratory mechanics resulting from increased diaphragm contractility and an increase in the respiratory rate, as demonstrated in this study.

Other studies have shown increased diaphragmatic electrical activity after caffeine loading [15–17]. However, these studies had small sample sizes, and Kraaijenga et al. [15] studied only spontaneously breathing newborns, monitoring diaphragmatic electrical activity by transcutaneous electromyography, a technique that does not allow for separate evaluation of diaphragmatic electrical activity during the inspiratory and expiratory phases. Furthermore, an increase in Edi measured by transcutaneous electromyography may partly result from other adjacent muscles, such as intercostal or abdominal muscles. Williams et al. [17] also evaluated the diaphragmatic response via electromyography and reported an increase in electrical activity and respiratory mechanics.

Kraaijenga et al. [15] expressed diaphragmatic activity as the percentage change in electromyography of the diaphragm (dEMG) and reported a 43% increase in dEMG after caffeine administration. In our study, we showed a 68% increase in the inspiratory peak Edi after the loading dose of caffeine. The comparison between the two studies revealed a greater increase in the inspiratory peak Edi when a more refined methodology was used.

Parikka et al. [16] reported that caffeine citrate reduced 5- to 10-second respiratory pauses. We found a reduction in the absolute number of respiratory pauses, which, however, was not statistically significant.

This study has several limitations. The use of a convenience sample diminishes its power; however, we did not find any literature reference using the same variables for sample size calculation. Since we found a significant difference in pre- and postcaffeine use in diaphragm electrical activity, we did not commit a type 2 error.

We could also not correlate the serum level of caffeine with diaphragmatic electrical activity because the caffeine serum level was collected together with other routine tests of newborns. Although we could not make this correlation, the literature shows that the standard doses commonly used provide the expected clinical effects, such as a reduction in apnea episodes and a reduction in extubation failures [32–34].

The absence of a control group in our study could be considered a limitation. However, from an ethical point of view, it would be inappropriate to postpone or stop administering a medication that is formally indicated for this group of newborns. In this case, each patient served as their own control through measurements taken before and after caffeine administration.

The duration of data recording and the possibility that the increase in Edi parameters may have been transient could be considered another limitation of our study. Kraaijenga et al. [15] and Williams et al. [17] reported that differences in diaphragmatic parameters were no longer significant beyond 120 min and 60 min from the caffeine load, respectively. However, Williams et al. studied only invasively ventilated newborns, and this respiratory support could lead to a decreased response of the diaphragm to caffeine stimulation. In our study, we could not correlate the ventilatory mode or lack of support with diaphragmatic electrical activity, as few patients were on mechanical ventilation (only 13.9%).

In preterm newborns, caffeine was effective at reducing apnea, reducing the need for invasive mechanical ventilation and lowering the rates of bronchopulmonary dysplasia, intracranial hemorrhage, and patent ductus arteriosus. Additionally, it improves the prognosis for pulmonary function and neuropsychomotor development [35]; its use is routine in neonatal intensive care units, and understanding its mechanism of action is extremely important for ensuring its safe use.

In conclusion, we demonstrated an increase in the electrical activity of the diaphragm in all phases of the inspiratory and expiratory cycles after the caffeine loading dose was administered to newborns, and adequate serum levels were observed. Additionally, we showed that the neural respiratory rate increased after caffeine loading, possibly reflecting a reduction in episodes of apnea.

## Author contributions

**Conceptualization:** Tatiana B.B. Zidan, Eduardo J. Troster, Jennifer Beck, Romy S.B. Zacharias, Celso M. Rebello.

**Data curation:** Tatiana B.B. Zidan, Celso M. Rebello.

**Investigation:** Tatiana B.B. Zidan, Celso M. Rebello.

**Methodology:** Tatiana B.B. Zidan, Livia R. Sanches, Carlos Eduardo S. Ferreira, Celso M. Rebello.

**Project administration:** Tatiana B.B. Zidan, Celso M. Rebello.

**Resources:** Livia R. Sanches, Carlos Eduardo S. Ferreira.

**Software:** Jennifer Beck.

**Supervision:** Tatiana B.B. Zidan, Eduardo J. Troster, Celso M. Rebello.

**Validation:** Tatiana B.B. Zidan.

**Visualization:** Tatiana B.B. Zidan, Celso M. Rebello.

**Writing – original draft:** Tatiana B.B. Zidan, Celso M. Rebello.

**Writing – review & editing:** Tatiana B.B. Zidan, Eduardo J. Troster, Jennifer Beck, Celso M. Rebello.

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
