## [Decision Letter · Decision Letter 0]

29 May 2024

PONE-D-24-08655Effect of caffeine citrate on diaphragmatic electrical activity in pre-term newbornsPLOS ONE

Dear Dr. Zidan,

Thank you for submitting your manuscript to PLOS ONE. After careful consideration, we feel that it has merit but does not fully meet PLOS ONE’s publication criteria as it currently stands. Therefore, we invite you to submit a revised version of the manuscript that addresses the points raised during the review process.

We look forward to receiving your revised manuscript.

Kind regards,

Hakan Aylanc

Academic Editor

PLOS ONE

Journal Requirements:

3. We note that you have a patent relating to material pertinent to this article. Please provide an amended statement of Competing Interests to declare this patent (with details including name and number), along with any other relevant declarations relating to employment, consultancy, patents, products in development or modified products etc. Please confirm that this does not alter your adherence to all PLOS ONE policies on sharing data and materials, as detailed online in our guide for authors http://journals.plos.org/plosone/s/competing-interests by including the following statement: ""This does not alter our adherence to  PLOS ONE policies on sharing data and materials.” If there are restrictions on sharing of data and/or materials, please state these. Please note that we cannot proceed with consideration of your article until this information has been declared.

**Additional Editor Comments:**

Dear writer,

Although the subject of the study is interesting, it will be re-evaluated if you respond adequately to the reviewer criticisms. You are expected to provide adequate explanations, especially regarding the availability of data and financial matters.

Reviewers' comments:

Reviewer's Responses to Questions

**Comments to the Author**

1. Is the manuscript technically sound, and do the data support the conclusions?

Reviewer #1: Yes

Reviewer #2: Partly

Reviewer #3: Yes

2. Has the statistical analysis been performed appropriately and rigorously? 

Reviewer #1: Yes

Reviewer #2: I Don't Know

Reviewer #3: Yes

3. Have the authors made all data underlying the findings in their manuscript fully available?

Reviewer #1: Yes

Reviewer #2: Yes

Reviewer #3: No

4. Is the manuscript presented in an intelligible fashion and written in standard English?

Reviewer #1: Yes

Reviewer #2: No

Reviewer #3: No

5. Review Comments to the Author

Reviewer #1: I congratulate the authors on conducting a "basic science” research; it is indeed essential to demonstrate clinically the pharmacological effects of commonly used drugs. I do have some suggestions

1, The fact that the sample size is a convenience sample makes it less powerful - it needs to be highlighted. I wonder why the authors could not have planned a presumed difference in findings before and after caffeine for a single primary outcome and calculation could have ben done. it would have increased the power of the study

2. where does the funding for the Edi probe (NAVA) probe come from when it is used for spontaneously breathing / those who did not require any supports? I am curious to know

3. The figures 1,2 are quite redundant-either the study flow diagram can be expanded to include the primary outcome or the figure can be dropped as such

the figure 2 also does not add any corroborative information to the methodology

4. Please explain why the signals could not be have been measured and recorded for analysis for a longer duration (like midway between 2 doses ) or when the trough level is expected (before the next scheduled dose)?

the probe should have been in situ or was it removed every time after measurement at 60 min post dose?

5. Table 1 can be better formatted- the max and min need not have a separate column only for 2 variables

6. Other details of the respiratory support like how many days of CPAP/NIV/IMV would have more relevance than the proportion of neonates needing each alone

also clinically relevant outcomes like BPD- to know the sickness levels of the babies- effect of caffeine and diaphragmatic activity would depend on secondary causes of apnea too.

Reviewer #2: In this study by Brasil Zidan et. al. evaluates the effect of a single dose of caffeine citrate on the electrical activity of the diafragm muscle. Authors show a relation between the EDI parameters before and after the first dose of caffeine citrate and conclude that caffeine in preterm infants promotes electrical activity of the diaphragm 60-120min after the first loading dose.

The design of the study is valid, yet minimalistic. As such it adds to our current understanding of the effects of caffeine on the electrical activity of the diafraghm. However, I have several remarks regarding the manuscript I would like to mention.

1. There are paragraphs of text that are duplicated in the text, as well as spelling errors (e.g. diaphrgm in the keywords)

2. The data is fully available on Google Drive. I would have like more information how this would be accesible.

3. The abstract line 37-39 is a very general statement about significance. What I would like to read is what the clinical effect is, not just if it is significant.

4. Abstract line 41, you mention 60-120 minutes after loading dose, but measurements were taken until 60min after the loading dose? (line 78) I believe it has something to do with the start of the recording (line 144), but it is confusing.

5. The authors mention the beneficial effect of caffeine and the fact that it increases diaphragmatic electrical activity (lines 65, reference 12). In line 76 the authors mention other studies investigating the electrical activity, I presume reference 12 should be added to the list as well? Taken together this study does not add significant new data to the already existing amount of data regarding electrical activity after caffeine. It would certainly strengthen the study design to include a much larger group of participants who are (especially) followed for a longer duration. As authors rightfully mention the the discussion. This would really expend the current knowledge about caffeine.

6. One of my main concerns is the patient groups. It includes many patients on mechanical ventilation, who are often sedated. Also how does one detect apnea's when on mechanical ventilation? As usually there is a set backup breathing frequency? Also many patients are included very early on after birth, where as many patients will develop apneu's within the first days after birth when spontaniously breathing. Suggesting a selection bias in this cohort.

7. Why are patient with HFO ventilation excluded?

8. statistics are mostly based on normality as is suggested by the representation as mean +- SD. In these small populations I think that a non-normal distribution is perhaps a better choice?

9. Line 192: sample calculation was not performed in my opinion, authors did use "sampling based on convenience" or "convenience sampling".

10. Line 257: I am unsure what the relationship is between de EDI values and the statement in this line. "These data" suggest that the EDI data is important for this statement, which it is not.

11. Paragraph from line 292 misses a conclusion of how these results should be interpreted in light of the current evidence.

12. Line 298, this section should also adress the question whether serum caffeine levels correlate with the clinical effect of caffeine, known in literature.

13. Line 309, in this section authors discuss the negative effect of mechanical ventilation on diaphragm function. This conclusion leads to a limition of this study (the fact that many patients were on MV) which should be discussed.

Minor:

line 214 missed dot.

line 259 the use of "methylxantines" is the first mentioning of this synonym for caffeine, it is confusing to start using at this point in the article.

Line 263: spelling error funtion

Line 266: I would like to know why this is beneficial?

In conclusion this study would greatly benefit from extensive editing and correction. It woulf further benefit from not only the electrical data but the subsequent effect on clinical parameters, which is a serious deficit in this study.

Reviewer #3: I would like to thank PLOS ONE for giving me the opportunity to review this interesting manuscript. By investigating the effect of a loading dose of caffeine citrate on diaphragmatic activity and neural respiratory rate in preterm neonates, the authors picked up an important topic for clinical neonatology. Based on the criteria for publication in PLOS ONE, here are my conclusions:

- The study definitively presents the results of primary scientific research.

- The presented results have not been published elsewhere.

- From my point of view, experiments and statistics were performed to a high technical standard and are described in sufficient detail.

- Conclusions are presented in appropriate fashion and are supported by the data provided.

- In general, the manuscript is presented in an intelligible fashion, using appropriate English. However, it would benefit from additional editing and I have summarized some corrections in my comments below.

- I have one issue regarding the required “research integrity”, and this is the registration in a clinical trials registry. I am actually worried that the authors state that they are still waiting for approval from the Brazilian Registry of Clinical Trials, although they are presenting a finished study which was performed from 2018 to 2019! Usually a clinical study should be registered before it is being performed, and it worries me that the authors obviously did not wait for the study registration to be completed. In addition, the authors state in their ethical considerations section that they did not aim for registration in a clinical trials system due to the observational character of the study, which by the way is no reason not to register (see https://clinicaltrials.gov/about-site/about-ctg#q2) – this makes me wonder even more why approval from the Brazilian Registry of Clinical Trials was ultimately sought?

- Regarding required data availability: The statement “Data is avaiable on Google Drive” does not specifically tell how to access data– please add a link or equal, unless you decided to do so after acceptance of the manuscript.

- Regarding adherence to appropriate reporting guidelines: I strongly encourage the authors to add a STROBE statement (https://www.equator-network.org/reporting-guidelines/strobe/) to their submission.

Furthermore, I have the following recommendations:

1) Abstract, lines 34-35: The presented numbers of females and males does not add up to 36. Furthermore, the patient numbers provided differ from the ones presented in the results section. Please correct this diligently!

2) I suggest chancing the manuscript’s short title to “Caffeine citrate and diaphragmatic electrical activity”, as this probably makes more sense.

3) Abstract, lines 36-37: I suggest adding the information when caffeine citrate levels were obtained.

4) Inclusion criteria, page 6, line 120: Please change “SMIV” to “SIMV”.

5) Evaluation of the electrical activity of the diaphragm, page 6, line 135: Please put “Maquet, Getinge Group, Solna, Sweden” in brackets or add “by” before this information.

6) Fig. 4, page 8, line 174: Please change “Minimun” to “Minimum”.

7) Results, pages 10-11: Please be consistent and present data with decimal numbers using a dot instead of a comma.

8) Results, page 11, lines 234-237: Please be consistent with the use of abbreviations – some, such as “MinIEdi”, “MaxIEdi” and “MinEEdi”, are introduced here, although these terms have been used before in the manuscript. This makes it difficult for the reader to follow the text.

9) Results, page 11, line 241 and Table 2: What does “mpm” stand for? I am not familiar with this abbreviation.

10) Discussion, page 12, line 261: I suggest changing “stimulate” to “stimulates”; otherwise, one could get the impression that adenosine stimulates central respiratory centers.

11) Discussion, page 13, lines 287-291: This paragraph is identical with the one at the top of the next page – please remove it.

12) Discussion, page 13, lines 296-297: I disagree with this sentence, as the authors only found a reduction in absolute numbers of respiratory pauses, which, however, was not statistically significant.

13) The reference list uses different styles of referencing, which should be harmonized using one standard referencing style. Furthermore, I have recognized several mistakes in writing, e.g. for reference 5 “Jonhson … Neo- natal …”, which also need careful correction.

14) References, page 17: Reference number 19 should actually be number 20 – please correct this.

6. PLOS authors have the option to publish the peer review history of their article (what does this mean? ). If published, this will include your full peer review and any attached files.

**Do you want your identity to be public for this peer review?** For information about this choice, including consent withdrawal, please see our Privacy Policy .

Reviewer #1: No

Reviewer #2: No

Reviewer #3: No

---

## [Author Response · Author response to Decision Letter 0]

8 Sep 2024

Here are our responses and the modifications made according to the suggestions.

Editor#:

1. Please ensure that your manuscript meets PLOS ONE's style requirem

ents, including those for file naming. The PLOS ONE style templates can be found at

We have revised the manuscript in accordance with the PLOS ONE guidelines.

The information about the consent form is included in the text, under Materials and Methods, as follows:

“Data were collected after obtaining informed consent (verbal and written) from parents”.

3. We note that you have a patent relating to material pertinent to this article. Please provide an amended statement of Competing Interests to declare this patent (with details including name and number), along with any other relevant declarations relating to employment, consultancy, patents, products in development or modified products etc. Please confirm that this does not alter your adherence to all PLOS ONE policies on sharing data and materials, as detailed online in our guide for authors http://journals.plos.org/plosone/s/competing-interests by including the following statement: ""This does not alter our adherence to PLOS ONE policies on sharing data and materials.” If there are restrictions on sharing of data and/or materials, please state these. Please note that we cannot proceed with consideration of your article until this information has been declared.

The study was developed and conducted at the Hospital Israelita Albert Einstein in São Paulo, and none of the authors involved have any conflicts of interest related to patents. The analysis of the collected data was performed in Toronto by Dr. Beck's team. We can confirm that Dr. Beck's participation in the study did not involve any conflicts of interest. However, for further clarification, we will provide a response to this inquiry as soon as possible.

We used the same abstract in the online submission and in the manuscript.

5. Please include captions for your Supporting Information files at the end of your manuscript, and update any in-text citations to match accordingly.

We have made corrections to the supporting information in accordance with the guidelines.

Reviwer 1#:

1. The fact that the sample size is a convenience sample makes it less powerful - it needs to be highlighted. I wonder why the authors could not have planned a presumed difference in findings before and after caffeine for a single primary outcome and calculation could have ben done. it would have increased the power of the study

The choice of a convenience sample was made because we were unable to conduct an appropriate sample size calculation, as we did not find any literature publications using the variables we employed in this study for such calculation. This was emphasized in the paper for the reader's better understanding.

“The use of a convenience sample diminishes its power; however, we did not find any literature reference using the same variables for sample size calculation. On the other hand, since we found a significant difference in pre- and post-caffeine use in diaphragm electrical activity, we did not commit a type 2 error”.

2. Where does the funding for the Edi probe (NAVA) probe come from when it is used for spontaneously breathing / those who did not require any supports? I am curious to know

The NAVA probes were acquired through a research grant from another study (FAPESP grant process number 2013/12499-0). We chose to acknowledge FAPESP (Fundação de Amparo à Pesquisa do Estado de São Paulo) in the acknowledgments and funding section.

3. The figures 1,2 are quite redundant-either the study flow diagram can be expanded to include the primary outcome or the figure can be dropped as such

the figure 2 also does not add any corroborative information to the methodology

Thank you for the suggestion. We have chosen to exclude Figure 1 and include the information in the text of the manuscript.

“Out of 78 eligible patients, 42 were excluded (4 major malformations, 18 refusal of consent, 7 data recording issues, 1 severe asphyxia and 12 concurrent use of the system), resulting in 36 included patients”.

4. Please explain why the signals could not be have been measured and recorded for analysis for a longer duration (like midway between 2 doses) or when the trough level is expected (before the next scheduled dose)?

the probe should have been in situ or was it removed every time after measurement at 60 min post dose?

The purpose of our study was to record the electrical activity of the diaphragm based on caffeine metabolism. It is kwown that the peak effect of caffeine citrate typically occurs about 30 to 60 minutes after intravenous or oral administration. This is the period when the concentration of caffeine in the blood reaches its highest point, providing the maximum stimulating effect. Based on these facts, the proposed methodology was to obtain the diaphragmatic electrical activity before drug administration and after its peak serum concentration.

Newborns who required an orogastric tube for feeding or newborns who were being ventilated using NAVA mode kept the probe in place. However, newborns who were able to receive oral feeding had the tube removed after measurements were taken. The behavior of caffeine levels and its impact on diaphragmatic electrical activity over time is valuable information and should be the subject of further studies. In our study, this was not possible because not all patients retained the probe after the caffeine loading dose.

5. Table 1 can be better formatted- the max and min need not have a separate column only for 2 variables

We appreciate the suggestion. The table has been formatted.

6. Other details of the respiratory support like how many days of CPAP/NIV/IMV would have more relevance than the proportion of neonates needing each alone

also clinically relevant outcomes like BPD- to know the sickness levels of the babies- effect of caffeine and diaphragmatic activity would depend on secondary causes of apnea too.

As the aim of our study was to evaluate diaphragmatic electrical activity after administration of the caffeine loading dose, most patients were within the first 48 hours of life. Therefore, it was not yet possible to correlate clinical outcomes such as ventilation time and diagnoses like bronchopulmonary dysplasia. We chose not to include further clinical information to avoid confusing the readers.

Reviwer 2#:

1. There are paragraphs of text that are duplicated in the text, as well as spelling errors (e.g. diaphrgm in the keywords)

Thank you for your observations. Errors have been corrected.

2. The data is fully available on Google Drive. I would have like more information how this would be accesible.

The analysis of diaphragmatic electrical activity data was performed by Dr. Jeniffer Beck's team at the University of Toronto, Canada. All communication was conducted through Google Drive, and if the editorial board of Plos One is interested, access can be granted through the following links:

https://drive.google.com/file/d/11i_Z9bctLByOly-wXb2qVssCeyJBTSTl/view?usp=drive_link

https://drive.google.com/file/d/1PAqt9EkuJRI5AQY0sCzssbHofqZ42kBt/view?usp=drive_link

3. The abstract line 37-39 is a very general statement about significance. What I would like to read is what the clinical effect is, not just if it is significant.

Thank you for your observation. We have changed as following.

“Results showed significant increases in all parameters used to determine the inspiratory phases of respiratory cycles post-caffeine administration (p < 0.001).”

4. Abstract line 41, you mention 60-120 minutes after loading dose, but measurements were taken until 60min after the loading dose? (line 78) I believe it has something to do with the start of the recording (line 144), but it is confusing.

Thank you for your observation. We agreed and have changed as following.

“The study concludes that caffeine administration significantly enhances diaphragmatic electrical activity in preterm newborns.”

5. The authors mention the beneficial effect of caffeine and the fact that it increases diaphragmatic electrical activity (lines 65, reference 12). In line 76 the authors mention other studies investigating the electrical activity, I presume reference 12 should be added to the list as well? Taken together this study does not add significant new data to the already existing amount of data regarding electrical activity after caffeine. It would certainly strengthen the study design to include a much larger group of participants who are (especially) followed for a longer duration. As authors rightfully mention the the discussion. This would really expend the current knowledge about caffeine.

The statement made in line 65 is incorrect and was well observed.

The cited study (reference 12) is an old study conducted on 6 adult patients that evaluated the diaphragm's response to caffeine through transdiaphragmatic pressure measurements obtained via esophageal catheters. According to this study, there was an increase in diaphragm contractility after oral administration of caffeine, and the electrical activity of the diaphragm was not evaluated.

Reference 15 is a study that evaluated diaphragmatic activity through transcutaneous electromyography, a different method from the one used in our study, which evaluated the electrical activity of the diaphragm through the NAVA catheter.

Reference 16 is a study that used the same methodology to obtain diaphragmatic electrical activity, but the main objective was to evaluate the occurrence of apneas after caffeine administration. A limitation of this study was the number of newborns studied, which was much smaller compared to our study.

Reference 17 is a study that used transcutaneous electromyography to evaluate the electrical activity of the diaphragm and only assessed newborns who were on mechanical ventilation.

6. One of my main concerns is the patient groups. It includes many patients on mechanical ventilation, who are often sedated. Also how does one detect apnea's when on mechanical ventilation? As usually there is a set backup breathing frequency? Also many patients are included very early on after birth, where as many patients will develop apneu's within the first days after birth when spontaniously breathing. Suggesting a selection bias in this cohort.

Newborns in our neonatal ICU, as well as in most ICUs, are not sedated when they are on mechanical ventilation. This allows them to maintain spontaneous breathing. None of the patients in our study were sedated. For further clarification for the reader, this comment was added to the text.

“Respiratory support was defined based on clinical parameters, with no changes between baseline and post-caffeine measurements. None of the pacients were sedated”.

Our study did not use clinical parameters to detect apneas during the monitoring of diaphragm activity. Respiratory pauses were evaluated through the absence of Edi for 5-10, 11-19, or more than 19 seconds. For this reason, there was no interference from mechanical ventilation in the newborns studied.

We do not agree that there is selection bias in the patients who used caffeine in this study because the inclusion criteria were based on protocols recommended in the literature. The objective of the study was not to select patients with apnea but to evaluate Edi secondary to the use of caffeine.

7. Why are patient with HFO ventilation excluded?

We were not sure if the Edi signal would be adequately transmitted, as there is chest vibration in newborns ventilated with high-frequency ventilation. In any case, there were no eligible cases for the study that used this ventilation mode during data collection.

8. Statistics are mostly based on normality as is suggested by the representation as mean +- SD. In these small populations I think that a non-normal distribution is perhaps a better choice?

As we described in Statistics, parameter distributions were studied using histograms and boxplots, and generalized estimating equation models were applied for the inferential analysis considering the correlation between measurements from the same newborn before and after caffeine administration. Normal or gamma distributions with a logarithmic link function were used, and the distribution that showed the best performance according to the QIC goodness-of-fit results was chosen.

9. Line 192: sample calculation was not performed in my opinion, authors did use "sampling based on convenience" or "convenience sampling".

The choice of a convenience sample was made because we were unable to conduct an appropriate sample size calculation, as we did not find any literature publications using the variables we employed in this study for such calculation. This was emphasized in the paper for the reader's better understanding.

“The use of a convenience sample diminishes its power; however, we did not find any literature reference using the same variables for sample size calculation. On the other hand, since we found a significant difference in pre- and post-caffeine use in diaphragm electrical activity, we did not commit a type 2 error”.

10. Line 257: I am unsure what the relationship is between de EDI values and the statement in this line. "These data" suggest that the EDI data is important for this statement, which it is not.

We agree that the statement is not clear because our data do not allow us to relate Edi to serum caffeine levels. To avoid errors in conclusions, we chose to exclude this sentence.

11. Paragraph from line 292 misses a conclusion of how these results should be interpreted in light of the current evidence.

We agree that the paragraph needs a conclusion. We added it to the text.

“The comparison between the two studies shows a greater increase in Edi using a more refined methodology”.

12. Line 298, this section should also adress the question whether serum caffeine levels correlate with the clinical effect of caffeine, known in literature.

We agree with the observation and add data from the literature to the text.

“ Although we could not make this correlation, the literature shows that the standard doses commonly used provide expected clinical effects such as: reduction in apnea episodes and reduction in extubation failures, without increasing the risk of neurodevelopmental disabilities”.

13. Line 309, in this section authors discuss the negative effect of mechanical ventilation on diaphragm function. This conclusion leads to a limition of this study (the fact that many patients were on MV) which should be discussed.

In our study we were unable to correlate the negative effects of mechanical ventilation on Edi as few patients were in this group (only 13.9%). If there was any effect it was probably small.

“In our study we could not correlate the ventilatory mode or lack of support with diaphragmatic electrical activity as few patients were on mechanical ventialtion (only 13,9%).”

Reviwer 3#:

1.I have one issue regarding the required “research integrity”, and this is the registration in a clinical trials registry.

The study was approved by the Ethics and Research Committee of our institution. This Committee is certified by AAHRPP (Association for the Accreditation of Human Research Protection Programs). Therefore, the ethical approval of our studies is

---

## [Decision Letter · Decision Letter 1]

2 Dec 2024

PONE-D-24-08655R1Effect of caffeine citrate on diaphragmatic electrical activity in pre-term newbornsPLOS ONE

Dear Dr. Zidan,

Thank you for submitting your manuscript to PLOS ONE. After careful consideration, we feel that it has merit but does not fully meet PLOS ONE’s publication criteria as it currently stands. Therefore, we invite you to submit a revised version of the manuscript that addresses the points raised during the review process.

We look forward to receiving your revised manuscript.

Kind regards,

Hakan Aylanc

Academic Editor

PLOS ONE

Journal Requirements:

**Additional Editor Comments:**

Although the improvements have been made are appropriate, in addition to the criticisms of the reviewer, I would like to make the following suggestions.

1. Spelling mistakes should be corrected with the support of a native English-speaking advisor.

2. There should be harmony between the abstract and the tables in the article, spelling mistakes should be corrected

3. 68% increase in peak inspiratory Edi”, according to Table 1, the maximum inspiratory Edi increased from 10.35 to 12.17, which corresponds to an increase of only 17.6%. Please explain.

4. The reference list should be corrected according to the spelling rules.

5. It is indicated that it was approved by the ethics committee. The full name of the ethics committee and the approval number, if available, should be included.

6. In lines 245,246, the conclusion of the study alone gives classical information and it would be good to combine the sentence with the sentence in 247 and 248.

Reviewers' comments:

Reviewer's Responses to Questions

**Comments to the Author**

1. If the authors have adequately addressed your comments raised in a previous round of review and you feel that this manuscript is now acceptable for publication, you may indicate that here to bypass the “Comments to the Author” section, enter your conflict of interest statement in the “Confidential to Editor” section, and submit your "Accept" recommendation.

Reviewer #3: (No Response)

2. Is the manuscript technically sound, and do the data support the conclusions?

Reviewer #3: Yes

3. Has the statistical analysis been performed appropriately and rigorously? 

Reviewer #3: Yes

4. Have the authors made all data underlying the findings in their manuscript fully available?

Reviewer #3: Yes

5. Is the manuscript presented in an intelligible fashion and written in standard English?

Reviewer #3: No

6. Review Comments to the Author

Reviewer #3: The authors have answered and addressed many of the questions and issues that arose during the first round of reviews, and I would like to thank them for their effort to improve their manuscript.

My main concern was the fact that approval from the Brazilian Registry of Clinical Trials was sought, but obviously not obtained. The argument that registration with ClinicalTrials.gov was not performed because “it was not an interventional study” is not valid, because ClinicalTrials.gov explicitly invites observational studies “in which researchers simply collect information”. Therefore, I am interested what the answer from the Brazilian Registry of Clinical Trials actually was?

While many of the implemented changes have helped improving the presentation of the manuscript, there are still some changes and improvements required:

1) Introduction, page 3, line 52: Please be consistent when using abbreviations such as AOP.

2) Patient population, page 6, line 115: Please change “pacients” to “patients”.

3) Table 1: There are differences in the presented mean birth weight between the abstract and the table – please check this carefully and correct it appropriately.

4) Results, page 10, line 221: I suggest changing “age of data collection” to “age at data collection”.

5) Table 2 and throughout: Please be consistent with the use of dots instead of commas for decimal numbers.

6) Discussion, page 12, line 245: Please change “demonstreted” to “demonstrated”.

7) Discussion, page 13, line 280: Where does the “68% increase in inspiratory peak Edi” come from? According to Table 1, the maximum inspiratory Edi increased from 10.35 to 12.17, which only corresponds to an increase of 17.6%. Please explain.

8) Discussion, page 13, lines 291-292: I suggest changing “… in the same moment that other routine testes of newborns” to “… together with other routine blood tests”.

9) Discussion, page 14, lines 294-295: The information about “without increasing the risk of neurodevelopmental disabilities” seems not relevant here and could be omitted.

10) Discussion, page 14, line 308: Please change “ventialtion” to “ventilation”.

11) The reference list has still not been comprehensively revised and corrected, e.g. for reference 5 “Jonhson” is still incorrect. Furthermore, there are still different styles of referencing used, such as in references 28 and 31.

7. PLOS authors have the option to publish the peer review history of their article (what does this mean? ). If published, this will include your full peer review and any attached files.

**Do you want your identity to be public for this peer review?** For information about this choice, including consent withdrawal, please see our Privacy Policy .

Reviewer #3: No

---

## [Author Response · Author response to Decision Letter 1]

3 Feb 2025

All responses to comments are in the attached Response to reviewers document.

---

## [Editor Report · Decision Letter 2]

28 Feb 2025

Effect of caffeine citrate on diaphragmatic electrical activity in pre-term newborns

PONE-D-24-08655R2

Dear Dr. Zidan,

We’re pleased to inform you that your manuscript has been judged scientifically suitable for publication and will be formally accepted for publication once it meets all outstanding technical requirements.

Kind regards,

Hakan Aylanc

Academic Editor

PLOS ONE
---

## [Editor Report · Acceptance letter]

PONE-D-24-08655R2

PLOS ONE

Dear Dr. Zidan,

I'm pleased to inform you that your manuscript has been deemed suitable for publication in PLOS ONE. Congratulations! Your manuscript is now being handed over to our production team.

Kind regards,

on behalf of

Dr. Hakan Aylanc

Academic Editor

PLOS ONE